# SeqMatchNet: Contrastive Learning with Sequence Matching for Place Recognition & Relocalization

**Sourav Garg**
QUT, Australia
s.garg@qut.edu.au

**Madhu Vankadari**
University of Oxford, UK

**Michael Milford**
QUT, Australia

**Abstract:** Visual Place Recognition (VPR) for mobile robot global relocalization is a well-studied problem, where contrastive learning based representation training methods have led to state-of-the-art performance. However, these methods are mainly designed for single image based VPR, where sequential information, which is ubiquitous in robotics, is only used as a post-processing step for filtering single image match scores, but is never used to guide the representation learning process itself. In this work, for the first time, we bridge the gap between single image representation learning and sequence matching through *SeqMatchNet* which transforms the single image descriptors such that they become more responsive to the sequence matching metric. We propose a novel triplet loss formulation where the distance metric is based on *sequence matching*, that is, the aggregation of temporal order-based Euclidean distances computed using single images. We use the same metric for mining negatives online during the training which helps the optimization process by selecting appropriate positives and harder negatives. To overcome the computational overhead of sequence matching for negative mining, we propose a 2D convolution based formulation of sequence matching for efficiently aggregating distances within a distance matrix computed using single images. We show that our proposed method achieves consistent gains in performance as demonstrated on four benchmark datasets. Source code available at https://github.com/oravus/SeqMatchNet.

**Keywords:** Visual Place Recognition, Localization, Contrastive Learning

## 1 Introduction

In the context of mobile robotics, Visual Place Recognition (VPR) is defined as the task of recognizing previously seen places by matching currently observed images with those stored in the reference map [1, 2, 3]. It is one of the key capabilities of a mobile robot and is required for the task of visual localization, either in the form of coarse location estimation [4, 5] given a prior map or for loop closures and mapping in visual SLAM [6, 7]. These VPR-driven tasks are relevant for many practical robotic applications, including driverless cars, last mile delivery and exploratory rovers.

One of the main challenges in VPR is scene appearance variations as a place can be revisited during a different time of day, season or weather conditions, for example, nighttime [8] or snowy conditions [9]. Under such difficult scenarios, classical feature encoding techniques often fail [10, 11, 12, 13]. Hence, learning-based techniques have been developed for both local [14, 15] and global feature representations [4, 16, 17]. In VPR, learning a single image descriptor has been accomplished by both classification [17, 18] and contrastive representation learning [4, 19] type approaches. While these single image based methods address the appearance challenges to some extent, dealing with significant perceptual aliasing typically requires the use of additional information in the form of image sequences [20, 21, 22, 23, 24].

In the VPR literature, several sequence-based models have been explored including the use of DTW [25], linear diagonal search [21], graphs [23] and particle filters [26]. Most of these methods follow an underlying common principle of aggregating match scores obtained by comparing single image representations. The community has thus focused on both the aspects, that

is, learning more robust single image global descriptors through a variety of deep learning techniques [4, 16, 19, 17, 27, 28] and developing robust sequence matching techniques [29, 30, 31, 32] – *but in isolation*. That is, single image representations are learnt without ever considering the subsequent task of sequence score aggregation (or sequence matching). Thus, there remains a disconnect between the two, as the latter never guides the learning process of the former. In this paper, we address this issue through triplet-loss based contrastive learning with the following key contributions:

- for the first time, we integrate sequence matching in the VPR training pipeline, bridging the gap between learning single image descriptors and sequence matching;

- we propose to use the sequence matching metric based on temporal order-preserved average of Euclidean distances for both computing triplet loss and mining negatives; and

- finally, to overcome the computational overhead of sequence matching for negative mining, we propose a novel 2D convolutions based distance matrix computation method.

We show that transforming an existing single image descriptor by directly optimizing it for sequence matching using the latter both as a loss function and for negative mining leads to superior performance, as compared to traditional formulations based on single image based distance metrics.

## 2   Related Work

A detailed literature review on VPR is provided in [1, 3, 33, 2]. Here, we cover the recent VPR literature involving learning-based and sequence-based techniques, and related literature that employs triplets-based contrastive learning techniques.

### 2.1   Learning for VPR

Recent advances in deep learning using CNNs have led to significant progress in the VPR literature, where superior performance was first achieved by simply using off-the-shelf deep-learnt image encoders [34, 35] for image retrieval [36] and place recognition [37], followed by analyses of different CNN layers and training data suitability [38, 39]. Several works exist that employ off-the-shelf CNNs in unique ways to improve VPR [40, 41, 42, 31, 43, 44, 45]. Learning directly for VPR, one of the most notable works, NetVLAD [4], re-established the well-known local feature aggregation technique: VLAD (Vector of Locally Aggregated Descriptor) [13] by employing a deep-learnt image encoder, a learnable VLAD layer and triplets-based contrastive learning. The follow-up literature has explored learning methods both in the context of feature aggregation and learning method types.

Chen et al. [17] trained AMOSNet and HybridNet on a place classification task to later use the learnt features as place representations. DeLF [46] and its recent global descriptor adaptation with DeLG [18] learnt features through a landmark recognition task. Sarlin et al. [27] proposed HF-Net employing knowledge distillation for simultaneously learning local and global features. Different pooling techniques such as GeM [16, 19], NetBoW [47, 48], NetFV [47], pyramid pooling [49, 50], max pooling [7], and graph attention pooling [51] have also been explored to learn robust image representations. Other learning based methods include descriptor standardization [52, 39], autoencoders [53], CapsuleNet [54] based feature segregation [55], and omnidirectional CNNs [56]. However, all these methods are specifically designed for single image based representation learning and are agnostic to any subsequent sequential processing.

### 2.2   Sequences for VPR

Use of sequential information in VPR is known to improve robustness against perceptual aliasing as more visual evidence becomes available to disambiguate false matches, which are otherwise hard to prevent through limited information or occasional transient noise within single images. Ho and Newman [20] proposed dynamic programming based sub-sequence searching to detect loop closures given a matrix of single image based distances. Milford and Wyeth [21] proposed SeqSLAM, a sequence searching VPR method based on local velocity search within a column-normalized score matrix. SMART [57] adapted SeqSLAM with the use of odometry information to deal with the velocity-sensitivity problem. Along the same lines, Lynen et al. [29] transformed the distance matrix to obtain "placial indices" and computed vote density using odometry information to form a "placeless" place recognition system. A number of follow-up works [22, 58, 23, 59, 60] have

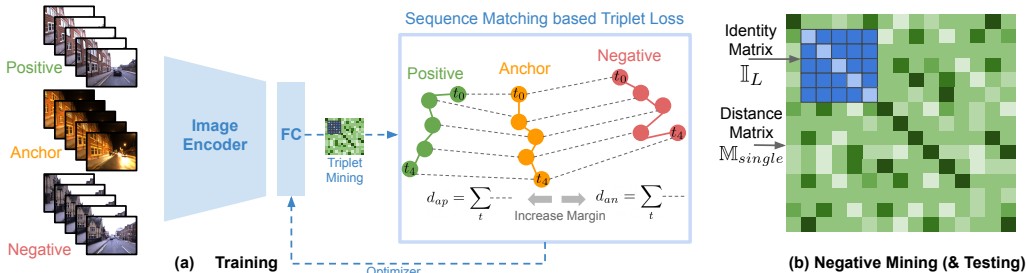

Figure 1: SeqMatchNet: (a) Images are encoded using an existing single image representation method and then a fully connected layer with bias is used as the learnable linear transform. The resultant *single* image descriptors are then processed as a sequence for computing triplet loss. Euclidean distance is only computed between *temporally corresponding* descriptors, say for the anchor and positive sequence, which is then averaged to obtain sequence matching distance $d_{ap}$. (b) For online negative mining, the distance matrix $\mathbb{M}_{single}$ (obtained from single image descriptor comparisons) is convolved with an Identity matrix $\mathbb{I}_L$ to efficiently compute sequence matching distances. Dashed blue lines show the training process.

proposed novel sequence searching techniques to robustly estimate place match hypotheses, for example, by using models based on Network Flow [22], HMM [61], Directed Acyclic Graph [23], kNN-DTW [25], and Particle Filter [26, 62, 32]. However, all of these methods are agnostic to the underlying feature representation techniques and do not use any learning-based mechanisms.

## 2.3   Triplet Loss and Contrastive Learning

Contrastive learning is used to learn a representation of the given signal by pulling closer the associated signals in contrast to other signals in the dataset [63, 64]. It has been employed for face verification [65], dimension reduction [66], object detection [67], video alignment [68], action recognition [69, 70], and several other tasks in scene understanding [71, 72, 73, 63, 74, 75]. Loss formulations typically differ among these methods; examples include noise-contrastive loss [76], normalized temperature based cross entropy loss [63], and max margin-based triplet loss [77] which is also used in this work. In the context of sequential imagery or videos, self-supervised triplet generation is quite common, as the positives can be sampled using either temporal proximity [78], 'clips' from within the same video [79, 80], another view of the same scene [81], or a different video from the same known cluster [82]. All of these methods aggregate frame-wise information explicitly by channel-wise concatenation of individual frames with 2D CNNs [70] or implicitly by encoding clips through a 3D CNN [79, 69, 80]. The supervisory signal in most of these methods is either based on strict timestamps [78] or nearby frames/clips [79, 82]. Contrary to that, *frame-level correspondences* are available for place recognition datasets through GPS or other sensors. This aids in defining supervised loss functions for accurate sequence alignment where order information can be explicitly enforced, unlike unordered set-based matching methods [83, 84, 85].

## 2.4   Learning with sequences for VPR

Recent work in VPR has started to explore learning based techniques for sequence-based methods but have mainly focused on generating sequential descriptors as a *single summary vector* representation of a sequence of images. This includes online discriminative learning of memory cells [86], end-to-end multi-view learning via feature grouping, fusion or recurrence [87, 88], learning descriptor grouping for sequence matching optimization [89], unsupervised coresets based summarization [90], topological map learning via recurrence [91], odometry/position based sequential learning [92, 93], unsupervised change-based descriptor adaptation using Delta Descriptors [94], and supervised 1D temporal convolution based hierarchical VPR with SeqNet [95]. In contrast to SeqNet [95] which aggregates single image *descriptors* within a sequence to generate a summary vector representation, the proposed SeqMatchNet aggregates single image based *match scores* given a pair of sequences. We do not perform any descriptor aggregation in this work but learn a linear transformation of single image descriptors, applied *independently* to each element within a sequence. Thus, sequential interaction is only introduced as the *sequence matching metric* when calculating loss and mining

negatives. This is in contrast to SeqNet which treats aggregated descriptors as a single vector for contrastive loss optimization.

## 3  SeqMatchNet

Here, we present all the components of *SeqMatchNet* (see Figure 1). We first explain how the existing single image representations are transformed; then we present our training framework which includes the triplet loss, sequence matching metric and convolutions-based distance matrix computation for negative mining; and finally, we describe the test time sequential query retrieval process.

### 3.1  Image Representation

Given an image $I$, its corresponding descriptor $X \in \mathbb{R}^D$ is assumed to be pre-computed using an existing single image representation method. This descriptor is linearly transformed to $X' \in \mathbb{R}^D$ using a learnable fully-connected layer with weight matrix $W \in \mathbb{R}^{D \times D}$ and bias $B \in \mathbb{R}^D$. In the supplementary material, we provide results for deeper network configurations.

$$X' = WX + B \tag{1}$$

$X'$ is l2-normalized before computing distances. $W$ and $B$ are shared across all the descriptors and also when computing triplets, as explained next.

### 3.2  Training

Here, we present the training pipeline including contrastive triplet loss and the use of sequence matching as a distance metric both for computing loss and mining negatives.

**Contrastive Triplet Loss:**   Our objective is to learn $X'$ such that for any given *anchor* image descriptor $X'_a$, its distance from a *positive* image descriptor $X'_p$ (belonging to a nearby physical location) is smaller than its distance from a *negative* image descriptor $X'_n$ (belonging to a far-off physical location) by a certain margin ($m$). That is, the contrastive max-margin triplet loss $\mathcal{L}$ is defined as:

$$\mathcal{L} = max(d_{ap} - d_{an} + m, 0) \tag{2}$$

**Sequence-based Distance Metric:**   In most existing works, the distance computation in the aforementioned triplet loss formulation is based on single summary vector metrics [4, 87]. In this work, we instead use a sequence of image descriptors of length $L$ centered at any given single image descriptor to define a distance metric, inspired by well-established sequence matching methods [21, 57, 23, 96, 95]. Thus, for a temporally-ordered list of images within a sequence, order-corresponding pairwise Euclidean distances are computed between any two sequences centered at $X'_i$ and $X'_j$, and then averaged to obtain the sequence matching distance[1] (see Figure 1):

$$d_{ij} = 1/L \sum_{t=-L/2}^{L/2} \|X'_{i+t} - X'_{j+t}\|_2 \tag{3}$$

The above sequential order-preserving distance measure is practical for many robotic applications, for example, autonomous driving, where an agent navigates repeatedly through the same route, especially when a shorter metric span (short sequence) is considered. Unlike several existing video-based learning methods [78, 79, 70, 69, 82], the availability of motion and position information (odometry/GPS) from robot's sensor data allows sub-sampling of image sequences using a fixed metric separation while also providing frame-to-frame correspondence across reference and query databases [6, 21, 4]. This enables simplified supervision with the help of the proposed distance metric and loss formulation for the task of visual place recognition, unlike unordered set-based matching metrics used in other domains [83, 84, 85].

Using sequence-based matching during training as proposed has a threefold effect. *1)* It *simultaneously* optimizes for $L$ consecutive descriptors – this prevents the system from over-focusing on

---

[1]$\mathbf{d}_{ij}$ follows all the distance metric properties since it is an average of an ordered list of Euclidean distances.

any one single descriptor within a sequence which is too hard (a positive or a negative) to optimize standalone (as in traditional approaches) but might only have a minimal impact on the aggregated sequence distance. This effect can be observed through the negative distance profiles visualized in Section 5.5. *2)* Since the individual descriptors within a sequence are matched in a temporal-order preserving manner, the anchor-positive distance so achieved is ideally the lowest – this prevents the system from unnecessarily optimizing for all possible combinations of anchors and positives within a certain localization radius, e.g. trying to minimize a full $L \times L$ distance matrix between an anchor and a positive sequence as opposed to ideally minimizing only the diagonal elements of that matrix. *3)* As an aggregated distance is computed between an anchor and a negative through sequence matching, the negatives selected online during training are harder than those which would have been selected when using single image based training – this gives priority to those regions of the environment where several consecutively-occurring negatives make it difficult to avoid perceptual aliasing.

**Distance Matrix Computation for Negative Mining:**  Matching sequences of image descriptors naturally comes at an additional cost of computation. This becomes a bottleneck during training when searching for hard negatives in a large reference database. Since most of the sequence-based VPR datasets are long vehicular trajectories of the environment, it is typical to first compute a distance matrix through single image descriptor comparisons. This distance matrix is then further processed to obtain sequence matching scores [20, 21, 5]. Leveraging GPUs' highly parallelized convolution operations, we propose an efficient 2D convolution based approach to compute sequence-based distance matrix $\mathbb{M}_{seq}$ that convolves an identity matrix $\mathbb{I}_L$ with the single descriptor-based distance matrix $\mathbb{M}_{single}$:

$$\mathbb{M}_{seq}(i,j) = \mathbb{I}_L * \mathbb{M}_{single}(i,j)/L \tag{4}$$

The above formulation computes the mean of the diagonal elements of $\mathbb{M}_{single}$ as $\mathbb{I}_L$ strides along its rows (reference database) and columns (query database), thus the outcome is the same as that of Equation 3. Using an identity matrix as a kernel enables a linear 1D warp between a given pair of image sequences, thus assuming uniform robot motion for both the sequences. However, Equation 4 can be easily adapted to deal with non-uniform motion by employing multiple binary kernels where 1s are determined by the warp to be searched such as the linear search proposed in SeqSLAM [21].

## 3.3  Testing

**Sequential Query Retrieval:**  During the test time, query images are retrieved using sequences of length $L$ centered at each query image. In the batch approach of sequence matching (Equation 4) for mining negatives during training, all the queries of the train set were available beforehand. For the test time, similar to an online operation, we assume query images to be accrued incrementally. Thus, we re-use Equation 4 for computing sequence matching distances but perform convolutions only along the rows of the matrix $\mathbb{M}_{single}$. Using the resultant distance matrix $\mathbb{M}_{seq}$, a reference image with the lowest sequence matching distance is retrieved as a match for any given query sequence.

# 4  Experimental Settings

## 4.1  Datasets

We use four datasets in this work: i) Nordland [9] - Summer vs Winter, ii) Oxford Robotcar [8] and iii) Brisbane City Loop [97] - Day vs Night, and iv) MSLS [88] - Multi-city/Multi-conditions. For the Oxford and Brisbane datasets, we subsample image frames at 2 meters for both training and testing. In the Nordland dataset, consecutive frames are approximately 20 meters apart. No subsampling is performed for the MSLS dataset. Additional details about the datasets and train/test splits are provided in the supplementary material.

## 4.2  Pre-processing and Evaluation

We used NetVLAD with PCA+Whitening [4] as our image descriptor of size $D = 4096$, computed by downsampling images to a resolution of $640 \times 320$. The learnt descriptors are retained to be of the same size $D$ to enable fair comparison against vanilla NetVLAD baselines. For sequence matching,

$L = 5$ is used for both training and testing. For testing, localization radius of 10 meters, 20 meters and 10 frames is used respectively for Oxford, Brisbane/MSLS and Nordland dataset[2].

We use Recall@K [98] as the evaluation metric as also used by previous works [4, 95, 87, 99]. It is defined as the ratio of queries that are correctly matched within its top-K retrieved images to the total number of queries. Please refer to the supplementary material for training parameters details.

## 5  Results and Discussion

In this section, we first present the key results of improving VPR using sequence matching within the training pipeline both as a loss function and for mining negatives, which is compared against the vanilla method of using single image based distance metrics. Then, we present results for cross-city testing of models trained on different cities to observe generalization trends. Finally, we present results for state-of-the-art VPR comparisons considering a variety of sequence-based approaches along with an ablation study on the length of sequence used. In order to better understand the performance variations between single image and sequence-based methods, we present a study on cross-testing single/sequence matching with/without sequence-based training in Section 5.2 of the supplementary material.

Table 1: Ablation study of different loss types and mining types for sequence-based VPR using Recall@K (1/5/20).

|  | **Loss Type** | **Mining Type** | **Oxford** | **Nordland** | **Brisbane** |
|---|---|---|---|---|---|
| *Existing:* | Single | Single | 0.76/0.89/0.97 | 0.59/0.75/0.86 | 0.50/0.62/**0.75** |
| *Proposed:* | SeqMatch | Single | **0.79/0.90**/0.97 | 0.61/0.76/0.87 | **0.51**/0.62/0.74 |
|  | Single | SeqMatch | 0.76/0.89/0.97 | **0.66/0.81/0.91** | **0.51/0.63/0.75** |
|  | SeqMatch | SeqMatch | *0.78*/0.89/0.97 | **0.66/0.81/0.91** | **0.51/0.63/0.75** |

### 5.1  Sequence Matching based Loss and Negative Mining

Table 1 shows how the choice of matching metric affects the recall performance when defining the loss and mining negatives. For this, we consider Equation 3 for computing loss and negatives during training using a sequence length of 1 or 5, referred to as *Single* and *SeqMatch* respectively in Table 1. Results reported on the train/test splits of each of the three datasets are based on the use of sequence matching for test-time query retrieval. It can be observed that given the retrieval task of sequence matching, using SeqMatch as a metric during training for *both* loss and mining leads to superior performance for all the datasets (last row: SeqMatch-SeqMatch) as compared to when not using it at all (first row: Single-Single). Furthermore, when using a cross-combination (second and third row), use of SeqMatch either retains or improves performance. In particular, it can be observed that the Oxford dataset necessarily requires SeqMatch-based loss to improve performance, leading to an absolute increase in Recall@1 by 3%. On the other hand, the Nordland dataset benefits more from SeqMatch-based mining, leading to a 7% absolute performance improvement. This study shows the importance of optimizing single image descriptors for sequence matching-based VPR using SeqMatch as the metric (Equation 3) during training.

Table 2: Cross-City VPR Performance: Recall@K (1/5/20) when training on one city and testing on the other. Best is boldfaced and second best italicized per column.

| **Train City\Test City** | Oxford | Brisbane | MSLS-Amman |
|---|---|---|---|
| Vanilla NetVLAD + Seq. Matching | 0.67/0.79/0.90 | 0.21/0.27/0.37 | 0.25/0.34/*0.40* |
| Oxford | **0.78/0.89/0.97** | *0.29/0.38/0.50* | *0.26*/**0.36/0.41** |
| Brisbane | *0.70/0.84/0.94* | **0.51/0.63/0.75** | *0.26*/0.35/**0.41** |
| MSLS-Melbourne | 0.58/0.72/0.86 | 0.22/0.27/0.36 | **0.27**/*0.35/0.40* |

---

[2]It is common to use the Nordland dataset with units of localization radius as frames instead of meters [9, 98, 87, 95].

Table 3: Comparison against other approaches: Recall@1/5/20 with train/test across cities.

| | Methods | Oxford | Brisbane |
|---|---|---|---|
| *No Sequence:* | NetVLAD [4] | 0.47/0.70/0.85 | 0.20/0.28/0.41 |
| *Sequential Descriptor:* | Smoothing [94] | 0.59/0.72/0.85 | 0.20/0.25/0.32 |
| | Delta [94] | 0.37/0.55/0.74 | 0.20/0.33/0.50 |
| | SeqNet ($S_5$) [95] | **0.62/0.76/0.88** | ***0.32/0.40/0.55*** |
| *Sequence Matching:* | NetVLAD [4] | 0.67/0.79/0.90 | 0.21/0.27/0.37 |
| | $S_1$ [95] | ***0.71**/0.83/0.93* | 0.28/0.36/0.48 |
| | $S_1$ [95] + GISM [58] | 0.65/-/- | 0.26/-/- |
| | $S_1$ [95] + GRH [5] | 0.34/-/- | 0.18/-/- |
| | **Ours**: SeqMatchNet | 0.70/***0.84/0.94*** | **0.29/0.38/0.50** |
| *Hierarchical:* | HVPR ($S_5$, $S_1$) [95] | ***0.71**/0.82/0.88* | 0.29/***0.40/0.55*** |
| | **Ours**: HVPR ($S_5$, SeqMatchNet) | ***0.71**/0.82/0.88* | **0.30/*0.40/0.55*** |

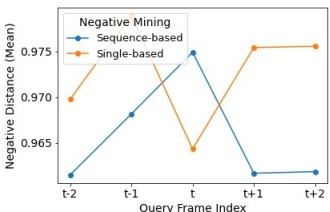

Figure 2: Query-negative distance profiles for single (orange) and sequence-based negative mining (blue).

Table 4: Effect of Sequence Length

| Method/Dataset | Oxford | | | Brisbane | | |
|---|---|---|---|---|---|---|
| | L=5 | L=10 | L=20 | L=5 | L=10 | L=20 |
| NV + Seq. Matching | 0.67/0.79/**0.90** | 0.79/0.86/0.93 | 0.92/0.94/0.96 | 0.21/0.27/0.37 | 0.24/0.28/0.37 | 0.26/0.30/0.40 |
| SeqNet ($S_5$) [95] | 0.62/0.76/0.88 | 0.64/0.75/0.88 | 0.58/0.71/0.85 | **0.32/0.40/0.55** | **0.33**/0.39/**0.52** | 0.32/0.37/0.47 |
| **Ours**: SeqMatchNet | **0.70/0.82**/0.88 | **0.84/0.92/0.96** | **0.94/0.97/0.99** | 0.29/0.38/0.50 | 0.31/**0.40/0.52** | **0.38/0.46/0.55** |

## 5.2 Cross-City VPR Performance

In Table 2, we present sequence matching results for training on one city and testing on the other. Even though the environmental conditions vary across different cities, cross-city generalization trends look promising. For example, SeqMatchNet-based Recall@1 for Oxford (0.70) and Brisbane (0.29) during cross-testing is significantly better than the use of sequence matching on top of vanilla NetVLAD which respectively achieves 0.67 and 0.21 (in the first row, referred to as *No Transformation*). A similar trend in relative performance improvements can be observed for MSLS-Amman, where trained models from all three cities lead to higher recall.

## 5.3 Benchmarking against State-of-the-Art Sequence-based Methods

Sequence-based information can be leveraged in different ways to improve single image based VPR performance. In Table 3, we compare against a variety of methods as following. 1) *Sequential descriptors* refer to single summary vector representation constructed through a sequence of images, which includes Smoothing/Averaging [94], Delta Descriptors [94] and SeqNet [95]. 2) *Sequence matching* refers to aggregation of single image based distances which can either be based on the diagonal of the matrix or graph-based path search within the matrix [58, 5]. For the former, we consider SeqNet's single frame version $S_1$ which is the same as using SeqMatchNet but with $L = 1$ in Equation 3, that is, without guiding single image descriptors with sequence matching metric. For the latter, we consider Graph-based Image Sequence Matcher [58], referred to as GISM, and Graph-based Relocalization using LSH [5], referred to as GRH. 3) *Hierarchical* sequence matching is considered as per [95] where sequential descriptors shortlist candidates for sequence matching. The first row in Table 3 does not use any sequential information and corresponds to single image based VPR baseline. For all other methods, we use a sequence length of $L = 5$. For all methods that use training, results are presented using models trained on the other city. The best results within any block are boldfaced and the overall best are additionally italicized. The underlying single image descriptor for all the methods is either NetVLAD [4] or an adaptation of NetVLAD as cited.

It can be observed that in most cases, use of sequential information improves performance over single image descriptors as expected. SeqMatchNet establishes a new state-of-the-art on the Oxford dataset considering Recall@K(5,20) and achieves the best results on the Brisbane dataset within the sequence matching block. Among the sequential descriptors, SeqNet [95] achieves state-of-the-art results on the Brisbane dataset but these results deteriorate when using $S_5$ within HVPR due to inferior sequence matching of $S_1$. However, this is not the case with the HVPR combination of $S_5$ and SeqMatchNet, where Recall@1 improves over the baseline HVPR while maintaining the

overall state-of-the-art Recall@K(5,20). This shows the relevance of improved sequence matching with SeqMatchNet which benefits from the inclusion of sequence matching in its training pipeline.

## 5.4 Effect of Sequence Length

Table 4 shows the effect of sequence length on the recall performance of the proposed method as compared to SeqNet [95] and vanilla NetVLAD with sequence matching. For both the datasets (tested using models trained on the other city), it can be observed that SeqNet's performance deteriorates with an increase in the sequence length. This can be attributed to its sequence average pooling layer which can potentially reduce the discriminative ability of the descriptors during retrieval when using longer sequences for such orderless averaging. On the other hand, in the proposed SeqMatchNet, averaging of distances between corresponding pairs of images within a sequence enforces an explicit order (as a linear identity warp between two short time-series), leading to state-of-the-art performance especially when longer sequence lengths are used. The proposed method also outperforms vanilla NetVLAD based sequence matching which shows that the learnt descriptor transformation through SeqMatchNet leads to robust representations suited to the task.

## 5.5 Visualizing Query-Negative Distance Profiles

In our proposed method, we use sequence matching to mine harder negatives than what is achievable through single descriptor matching. In Figure 2, we show the single image distances between a query sequence and its negatives (mean of top 50 hard negatives) *before* averaging distances within a sequence, where negatives are computed using either single descriptors (orange) or sequence-based matching (blue). It can be observed that single descriptors based matching (orange) leads to negative selection solely based on the central elements of the sequence at query frame index $t$. This is apparent from the dip in its distance profile at $t$ while disregarding the corresponding distance values in its vicinity. On the other hand, sequence-based negative mining (blue) simultaneously considers all the elements in the sequence and is also able to select negatives when the central element is not necessarily the hardest one but the overall sequence indeed is.

# 6 Conclusion

Recent advances in visual place recognition have been accelerated by learning based techniques, which are predominantly governed by single image based distance metrics. The use of sequential information, although well established in the literature, has never been explored in the context of biasing single image representations to suit the subsequent task of sequence matching. In this paper, we bridge this gap for the first time through *SeqMatchNet* and show that overall VPR performance can be improved by including sequence matching in the training pipeline by adapting an existing single image descriptor. For this, we proposed temporal order-preserved average of Euclidean distances as the sequence matching metric which is used to both compute the loss and mine negatives online. To reduce the overhead of sequence matching for the latter, we proposed a novel 2D convolution-based processing of the distance matrix. We established that the single image descriptors learnt through SeqMatchNet are more robust and lead to superior performance with the help of sequence matching metric as opposed to the use of single image matching metrics.

In mobile robotics, localization pipelines are typically comprised of several steps of robustness inducing processes. The same is true of VPR where single image global representation matching is followed by performance enhancement techniques like sequential processing [20, 21], local feature matching [100, 42], feature filtering [101, 102], query expansion [103], and even dimensionality reduction [4]. One can expect that including these subsequent processes in the representation learning process can improve the overall robustness. This work takes a step in that direction by the inclusion of sequence matching in the training itself which leads to consistent performance gains. This study will hopefully encourage the researchers to reconsider the existing VPR and localization pipelines which can benefit from end-to-end task-relevant learning.

**Acknowledgments**

We acknowledge the continued support from the QUT Centre for Robotics and thank the reviewers and the editor for their useful comments. We used Weights & Biases [104] for experiment tracking for this paper.

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
