# OpenReview forum: "SeqMatchNet: Contrastive Learning with Sequence Matching for Place Recognition & Relocalization"
_robot-learning.org/CoRL/2021/Conference — CoRL2021 Oral_

### Official Review · Reviewer_Kfyz · 2021-07-16

**Originality:** Good
**Technical Quality:** Good
**Clarity Of Presentation:** Good
**Impact:** 3

**Recommendation:**

Strong Accept: I recommend accepting the paper and will argue for my recommendation even if other reviewers hold a different opinion.

**Summary:**

The paper presents an approach for visual place recognition dubbed SeqMatchNet. The idea is to bridge the gap between sequence matching methods and robustifying image descriptors for visual place recognition. This is accomplished by introducing a contrastive triplet loss which instead of being computed on the descriptor of a single query image, is computed instead on the distance metric calculated on a sequence of image descriptors. In order to speed up computation, the distance metric can be efficiently computed by a convolution operation. The method is evaluated thoroughly on a number of datasets.

**Issues:**

See above.

**Reviewer Expertise:**

Fair: Some knowledge of the area

**Strengths And Weaknesses:**

The paper proposes a unique approach for bridging the gap between the different research directions in the area of visual place recognition, and demonstrates the utility of combining both a contrastive loss approach with sequence matching.

The list below should be viewed more as questions that are unclear to this reviewer rather than weaknesses of the approach:
1) For the sequence-based distance metric: one of the advantages claimed is that negatives mined during training are harder than those selected using single image based training. Can you elaborate more on why this is the case? Is there some empirical evidence supporting this claim in your findings?
2)  For table 1 and table 2, what is the baseline approach used? Is it a simple NetVlad model?
3) In line 251, the good performance is attributed to the importance of optimizing single image descriptors. Where in the presented approach are the image descriptors optimized?
4) For the cross-city VPR performance, it would be interesting to see the performance of other SOTA methods as well. Currently, it's hard to judge the performance of this approach without this comparison.
5) Finally a bit of a naive question, but how do the approaches generalize to a new dataset after being trained on a particular training scene? The dataset and the matching matrices need to be computed from scratch for every scene correct? So where does the generalization bit come in?

**Summary Of Recommendation:**

I do not feel strongly about accepting nor rejecting the paper without having the above questions clarified.

EDIT:

After the author's responses I feel more confident in arguing for the acceptance of the work.

---

> ### Author Response · Authors · 2021-08-31
> **Response to Reviewer 3 (Kfyz)**
>
> > *1. For the sequence-based distance metric: one of the advantages claimed is that negatives mined during training are harder than those selected using single image based training. Can you elaborate more on why this is the case? Is there some empirical evidence supporting this claim in your findings?*
>
> **R3.1** We have now included Section 5.5 and Figure 2 in the revised manuscript for visualizing query-negative distance profiles. In Figure 2, we can visualize the two cases: i) when negatives are selected using single images (orange) and ii) when negatives are selected using sequence matching (blue). In the mean negative distance plot on the left, it can be observed that single descriptor based matching (orange) leads to negative selection solely based on the central elements of the sequence at query frame index t. This is apparent from the dip in its distance profile at t while disregarding the corresponding distance values in its vicinity. Conversely, sequence-based negative mining (blue) simultaneously considers all the elements in the sequence and is consequently able to appropriately select negatives when the overall sequence is the correct negative, but the central element is not necessarily the hardest individual element.
>
>
> > *2. For table 1 and table 2, what is the baseline approach used? Is it a simple NetVlad model?*
>
> **R3.2** In Table 1 of the manuscript, all results correspond to the use of NetVLAD + learnt FC layer (from Equation 1) + Sequence Matching (during testing). As a baseline, the first row is the vanilla learnt version of the proposed system where the loss and negatives are computed without any use of sequential information during training. The other three rows ablate results for sequence-based loss and negative mining during training which show that the proposed learning technique leads to improved performance.
>
> In Table 2 of the revised manuscript, we have now updated the method description of the first row to be more clear - it corresponds to the use of vanilla NetVLAD + Sequence Matching as the baseline, that is, without any further learnt transformation on top of the vanilla descriptors. In the remaining rows of the table, we use NetVLAD + FC (from Equation 1) + Sequence Matching to demonstrate cross-city generalization.
>
>
> > *3. In line 251, the good performance is attributed to the importance of optimizing single image descriptors. Where in the presented approach are the image descriptors optimized?*
>
>
> **R3.3** In L236 (previously L251), with “optimizing single image descriptors”, we referred to the learnt linear transformation applied to the vanilla NetVLAD descriptors, as defined in Equation 1 and using the loss function defined in Equation 2.*
>
>
> > *4. For the cross-city VPR performance, it would be interesting to see the performance of other SOTA methods as well. Currently, it's hard to judge the performance of this approach without this comparison.*
>
> **R3.4** We do indeed compare against several sequence-based state-of-the-art methods, results for which are reported in Table 3. We observed that the method SeqNet performs closest to the proposed SeqMatchNet. In the newly included Table 4 in the revised manuscript, we now show that SeqMatchNet outperforms both SeqNet and sequence matching applied to vanilla NetVLAD descriptors.
>
> > *5. Finally a bit of a naive question, but how do the approaches generalize to a new dataset after being trained on a particular training scene? The dataset and the matching matrices need to be computed from scratch for every scene correct? So where does the generalization bit come in?*
>
> **R3.5** The weights, that is, the transformation matrix and bias in Equation 1, are trained on a particular dataset (e.g., Oxford - Day vs Night) and then used to compute descriptors for any new unseen dataset (e.g., Brisbane - Day vs Night). Thus, the method is expected to generalize across cities.

---

> > ### Comment · Reviewer_Kfyz · 2021-08-31
> > **Acknowledging clarifications**
> >
> > I would like to thank the authors for their helpful clarifications and additional experiments. They indeed were very helpful in clarifying my main concerns.

---

### Official Review · Reviewer_CkaF · 2021-07-23

**Originality:** Good
**Technical Quality:** Good
**Clarity Of Presentation:** Good
**Impact:** 3

**Recommendation:**

Weak Accept: I recommend accepting the paper, but will not argue for my recommendation if the majority of other reviewers have a different opinion.

**Summary:**

The paper proposes a sequence matching learning formulation for place recognition and relocalization. Instead of learning the image metric with single images, the authors use the distance between images of the sequences in contrast for learning the feature embedding. A negative mining method is also used to improve the training process. The final results on several commonly used benchmarks show the proposed method is very promising.

**Issues:**

I am willing to improve my rating if the authors can provide strong evidence that the loss can be used for deeper embedding networks. The additional study mentioned in the weaknesses is important to make the design convincing.

**Reviewer Expertise:**

Good: General knowledge of the area

**Strengths And Weaknesses:**

STRENGTHS

The paper considers the sequence information to learn image embedding for place recognition and relocalization. I agree that it is an important source of information to explore.

WEAKNESSES

The feature learning with single-layer network is a curious choice. Usually, deep networks can provide better embedding with SGD. Some exploration of the embedding network, especially depth, is important for understanding the loss.

The ablation study on different training and testing schemes will be very helpful for understanding the method. The variants are 1) training on sequence and test on single images. 2) training on single images and test with a sequence?

It is unclear whether the major improvement comes from negative mining or not.

Failure case analysis will also help understand the results.


**Summary Of Recommendation:**

Although I agree that learning from sequences can be better, there are still a lot of unknowns for the current experiments.

----

After reading author's feedback, my concerns are largely addressed. I changed my recommendation to weak accept.

---

> ### Author Response · Authors · 2021-08-31
> **Response to Reviewer 2 (CkaF) Part: 1/2**
>
> >*“The feature learning with single-layer network is a curious choice. Usually, deep networks can provide better embedding with SGD. Some exploration of the embedding network, especially depth, is important for understanding the loss.”*
>
> **R2.1** Thanks for the suggestion. To explore this issue, we conducted new experiments with alternative network configurations, including both an end-to-end training with image encoder and one based on pre-computed descriptors. This is now referred to in the revised manuscript in Section 3.1, and details are presented in Supplementary Section 5.1.
>
> 1. **End-to-end training**:
> We used AlexNet as the image encoder with global max pooling to obtain a 256-dimensional single image descriptor. We found that with and without the proposed sequence-matching based loss and negative mining, Recall@1 dropped from 49.3 to 47.6 respectively on the Oxford Robotcar dataset. This demonstrates that the relative performance gains of the proposed method are still achievable even when using an end-to-end training process with a deep architecture. We observed that the absolute performance is affected by the choice of architecture, pooling and the training dataset type; and that using precomputed descriptors achieves superior performance.
>
> 2. **Pre-computed Descriptors**:
> Supplementary Table 3 shows Recall@1 on the Oxford Robotcar dataset when using different network configurations for pre-computed single image descriptors - in this case, PCA-transformed 4096-dimensional NetVLAD vectors. It can be observed that the use of additional layers with or without the use of non-linearity does not further improve performance. This can be attributed to two specific aspects of the input NetVLAD descriptors: i) PCA-transformation, which can potentially facilitate learning of linear combination of features, and ii) high dimensionality (4096), which leads to a significant increase in the network size (number of learnable parameters) as more layers are included.
>
>
> >*“The ablation study on different training and testing schemes will be very helpful for understanding the method. The variants are 1) training on sequence and test on single images. 2) training on single images and test with a sequence?”*
>
> **R2.2** Thanks for the suggestion. We have now included Supplementary Section 5.2 and Supplementary Table 3 that investigate the effect of training with and without sequences on both single image and sequence-based matching. It can be observed that sequence matching based Recall@1 improvement is much higher for the trained systems as compared to the vanilla NetVLAD descriptors. Furthermore, using sequential information both during training and testing consistently leads to superior performance, which is closely followed by training on single and testing on sequence. Finally, for the Oxford dataset, it is interesting to note that recall of single descriptor testing reduces slightly due to training (0.46) as compared to the vanilla descriptors (0.47), even though using sequence matching on top of vanilla descriptors is significantly inferior to its trained counterparts.
>
>
> >*“It is unclear whether the major improvement comes from negative mining or not. Failure case analysis will also help understand the results.”*
>
> **R2.3** We refer the reviewer to Table 1 in the paper, where we ablate the single image-based and sequence-based mining of negatives. It can be observed that as compared to single image based negative mining (first row), sequence matching based mining (last two rows) consistently leads to improved performance. Consequently we can conclude that the performance improvement is coming from the sequence matching rather than negative mining.
>
> To further elaborate this, we have now included Section 5.5 and Figure 2 in the revised manuscript for visualizing query-negative distance profiles. In Figure 2, we can visualize the two cases: i) when negatives are selected using single images (orange) and ii) when negatives are selected using sequence matching (blue). In the mean negative distance plot on the left, it can be observed that single descriptors based matching (orange) leads to negative selection solely based on the central elements of the sequence at query frame index t. This is apparent from the dip in its distance profile at t while disregarding the corresponding distance values in its vicinity. On the other hand, sequence-based negative mining (blue) simultaneously considers all the elements in the sequence and is also able to select negatives when the central element is not necessarily the hardest one but the overall sequence indeed is.

---

> > ### Author Response · Authors · 2021-08-31
> > **Response to Reviewer 2 (CkaF) Part: 2/2**
> >
> > >*“I am willing to improve my rating if the authors can provide strong evidence that the loss can be used for deeper embedding networks. The additional study mentioned in the weaknesses is important to make the design convincing.”*
> >
> > **R2.4** In the newly included Supplementary Section 5.1, we show that the proposed loss function can indeed be used with deeper embedding networks. In the newly included Supplementary Section 5.2, we also investigate the effect of training with and without sequences on both single image and sequence-based matching.

---

### Official Review · Reviewer_A1mn · 2021-08-07

**Originality:** Very Good
**Technical Quality:** Very Good
**Clarity Of Presentation:** Very Good
**Impact:** 3

**Recommendation:**

Strong Accept: I recommend accepting the paper and will argue for my recommendation even if other reviewers hold a different opinion.

**Summary:**

The paper presents an integrated deep-based approach for visual place recognition (VPR) that combines sequence learning of feature descriptors with sequence matching. Particularly, a Euclidean distance metric is introduced in a triplet loss, to provide order-preservation of single image descriptors in the sequence. To facilitate the training, the authors also a propose a simple method for negative mining based on the Euclidean distance metric in the sequence data.

**Issues:**

While the paper is well-written and well-organized, my concern regard the similarities and differences between SeqNet and SeqMatchNet. It seems that the main component of the method is SeqNet with an extension to the triplet loss. This is of course fine, as long as a rigorous examination of the benefits of the one method over the other is demonstrated. Firstly, it would be interesting to ablate the length of the sequence, as only a single vs. Length of 5 is provided. Moreover, the authors claim state of the art results in Table.4 however the increase in recall are marginal or same as SeqNet. Is it possible to provide evidence for the significance of the prediction improvement?
I think this will also benefit the community regarding the research direction you are proposing (i.e., embedding sequence matching in the training process).

**Reviewer Expertise:**

Good: General knowledge of the area

**Strengths And Weaknesses:**

The paper has a very rigorous examination of the state of the art works. I find the idea of integrating the sequence matching into the training process of VPR reasonable, The paper is well-written and easy to follow. Adding an algorithmic block describing the training process would be helpful.
My only concern regards the significance of the provided results and the fundamental differences of the proposed work w.r.t. SeqNet, that uses learning of image descriptors in sequences and produces hypotheses on sequence matches, to be later on refined.

**Summary Of Recommendation:**

UPDATE: After rebuttal and the update of the paper with stronger ablations, I change my score to Strong accept, as I believe the paper bring significant improvement w.r.t. the prior works.

My recommendation is mainly driven by the concern described above, namely I am not convinced about the benefit of the method compared to SeqNet, as the result of table 3 does not seem significant.

---

> ### Author Response · Authors · 2021-08-31
> **Response to Reviewer 1 (A1mm)**
>
> >*“My only concern regards the significance of the provided results and the fundamental differences of the proposed work w.r.t. SeqNet, that uses learning of image descriptors in sequences and produces hypotheses on sequence matches, to be later on refined.”*
> >
> >*“my concern regard the similarities and differences between SeqNet and SeqMatchNet. “*
> >
> >*“It seems that the main component of the method is SeqNet with an extension to the triplet loss. This is of course fine, as long as a rigorous examination of the benefits of the one method over the other is demonstrated.”*
>
> **R1.1** Here, we describe the fundamental differences between SeqNet and SeqMatchNet in terms of the descriptor type and matching method type during training:
> 1. **Descriptor Type**:
> SeqNet uses temporal convolutions to aggregate single image “descriptors” within a sequence to generate a compact vector representation. On the other hand, the proposed method SeqMatchNet aggregates single image based “match scores” (Euclidean distance in this case) given a pair of sequences. We do not perform any descriptor aggregation in this work. The learnable linear transformation (Eq 1) is applied independently to the single image descriptors within a sequence, and it is only when the loss and negatives are calculated that we bring in the sequence information, as explained next.
> 2. **Matching Type During Training**:
> The prior SeqNet system learns an aggregated “sequential descriptor” which is treated as a single image descriptor for loss optimization. In contrast, the proposed method SeqMatchNet focuses on learning single image descriptor transformation which is optimized directly by the use of “sequence matching” for both computing loss and mining negatives.
>
> >*“My recommendation is mainly driven by the concern described above, namely I am not convinced about the benefit of the method compared to SeqNet, as the result of table 3 does not seem significant.”*
> >
> >*“Moreover, the authors claim state of the art results in Table.3 however the increase in recall are marginal or same as SeqNet. Is it possible to provide evidence for the significance of the prediction improvement? I think this will also benefit the community regarding the research direction you are proposing (i.e., embedding sequence matching in the training process).”*
>
> **R1.2** We have now included a new ablation study over different sequence lengths. This is presented in Section 5.4 of the revised manuscript along with the newly included Table 4. This new study shows that the proposed SeqMatchNet achieves state-of-the-art performance, and that its performance advantages become particularly significant when using longer sequence lengths. Longer sequence lengths are often required to successfully perform place recognition in particularly challenging circumstances and hence this result highlights the advantages of the new proposed SeqMatchNet system in such conditions.
>
> For the two datasets (tested using models trained on the other city), we show that SeqNet's performance deteriorates with an increase in the sequence length while the proposed SeqMatchNet is able to leverage longer sequences to find better matches. The proposed method also outperforms vanilla NetVLAD based sequence matching which shows that the learnt descriptor transformation through SeqMatchNet leads to robust representations suited to the task.
>
> >*“Adding an algorithmic block describing the training process would be helpful.”*
>
> **R1.3** To help better understand the training process, we have now updated the schematic in Figure 1 of the revised manuscript where the training process is highlighted with dashed blue lines and includes feature extraction, negative mining, and loss optimization.
>
> >*“Firstly, it would be interesting to ablate the length of the sequence, as only a single vs. Length of 5 is provided.”*
>
> **R1.4** We have now included Section 5.4 in the revised manuscript along with Table 4 to study the effect of varying sequence length on the recall performance. Please refer to response **R1.2**.

---

> > ### Comment · Reviewer_A1mn · 2021-08-31
> > **Acknowledging authors responses and revisions**
> >
> > Thank you very much for responding to my concerns, Your responses have now cleared out my doubts. I propose that you integrate your discussion on R1.1. regarding the fundamental differences between SeqNet and SeqMatchNet in the paper, so that you make stronger your case about the paper's contributions.

---

### Meta-Review · Area_Chair_MqMF · 2021-08-11

**Recommendation:** Accept (Oral)
**Confidence:** 4

**Metareview:**

The idea proposed by the authors has been reasonably well received by the reviewers. Combining sequential information and learning of image descriptors showed to improve the results w.r.t. other baselines. While existing time contrastive learning methods have relied on sequences of images, the idea of leveraging sensor information for obtaining frame-level correspondences during training is well thought and clean.
The paper has a good technical quality and it is clear to read.

However, as pointed out by the reviewers,  certain parts of the paper have caused confusion and require more support such as the case for mining negatives and their real benefit, deeper analyses including ablation studies on sequence lengths, design choices on the network depth, and its benefits in regards to SeqNet.

Post-rebuttal ============

Thank you for the clarifications and discussions. The reviewers have been in general pleased with the answers and many of the concerns have been solved in the updated manuscript.

---

> ### Author Response · Authors · 2021-08-31
> **Response to AC & Summary of changes**
>
> We have addressed all the concerns of the reviewers through extensive new experiments and analyses, which are presented in the revised manuscript and the supplementary material and briefly overviewed below:
>
> i) @R1, a new section (5.4) in the revised manuscript for an ablation study of sequence length, which now clearly shows the advantages of the proposed method over existing techniques (including SeqNet) especially when using longer sequence lengths;
>
> ii) @R2, a new section (Supp. 5.1) in the supplementary investigating deeper network configurations, which demonstrates that the proposed method does indeed work for deeper architectures both with end-to-end training and when using pre-computed descriptors;
>
> iii) @R2, a new section (Supp. 5.2) in the supplementary on cross-testing single/sequence matching with/without sequence-based training, which provides a better understanding of the performance variations between single image and sequence-based methods, and how training with sequence matching helps;
>
> iv) @R3,R2, a new section (5.5) in the revised manuscript visualizing query-negative distance profiles, which shows how the negative selection varies between single descriptor based methods and sequence matching based methods;
>
> v) @R1, a detailed discussion on the points of difference between SeqNet and the proposed SeqMatchNet that also highlights the significance of performance gains provided by the new SeqMatchNet system; and finally,
>
> vi) @R1, an updated illustration of the proposed method (Figure 1) showing the entire training process including feature extraction, negative mining, and loss optimization.
>
> Other major changes include moving the Section “Visualizing Distance Margins”, and “Dataset Details” from the original manuscript to the supplementary due to space limitations.

---

### Decision · Program_Chairs · 2021-09-13

**Decision:**

Accept (Oral)

**Comment:**

The idea proposed by the authors has been reasonably well received by the reviewers. Combining sequential information and learning of image descriptors showed to improve the results w.r.t. other baselines. While existing time contrastive learning methods have relied on sequences of images, the idea of leveraging sensor information for obtaining frame-level correspondences during training is well thought and clean.
The paper has a good technical quality and it is clear to read.

However, as pointed out by the reviewers,  certain parts of the paper have caused confusion and require more support such as the case for mining negatives and their real benefit, deeper analyses including ablation studies on sequence lengths, design choices on the network depth, and its benefits in regards to SeqNet.

Post-rebuttal ============

Thank you for the clarifications and discussions. The reviewers have been in general pleased with the answers and many of the concerns have been solved in the updated manuscript.